# Alternative Lengthening of Telomeres Is Rare in Canine Histiocytic Sarcoma

**DOI:** 10.3390/cancers15174214

**Published:** 2023-08-22

**Authors:** Theresa Kreilmeier-Berger, Heike Aupperle-Lellbach, Martin Reifinger, Nicolai Valentin Hörstke, Klaus Holzmann, Miriam Kleiter

**Affiliations:** 1Department for Companion Animals and Horses, University of Veterinary Medicine, 1210 Vienna, Austria; theresa.kreilmeier-berger@vetmeduni.ac.at; 2Laboklin GmbH & Co. KG, 97688 Bad Kissingen, Germany; aupperle@laboklin.com; 3Department of Pathobiology, University of Veterinary Medicine, 1210 Vienna, Austria; m.reifinger@gmx.net; 4Center for Cancer Research, Comprehensive Cancer Center, Medical University of Vienna, 1090 Vienna, Austria; nico.hoerstke@gmail.com

**Keywords:** C-circle assay, telomere, prevalence study, orphan tumor

## Abstract

**Simple Summary:**

Knowledge of telomere maintenance mechanisms (TMMs) in different tumor types is crucial for the development of TMM-specific diagnostics and therapies. Telomerase-independent alternative lengthening of telomeres (ALT) is used more frequently in human sarcoma subtypes than other cancers. However, ALT’s prevalence in the aggressive hematopoietic and orphan tumor type termed histiocytic sarcoma (HS) is unknown. The aim of our retrospective study was to assess the ALT activity in canine HS as a surrogate for rare human HS cases. Bernese mountain dogs (BMDs) have a breed predisposition for HS. We show in cohorts of a total of 63 dog patients from two centers, including a homogeneous population of 47 BMDs, that ALT was used only infrequently in the BMDs and not at all in the non-BMD patients here. We conclude that ALT is useful for a few cases only to target HS but can play a role as a resistance mechanism after targeting telomerase.

**Abstract:**

Cancer cells activate telomere maintenance mechanisms (TMMs) to overcome senescence and thus are targets for TMM-specific therapies. Telomerase-independent alternative lengthening of telomeres (ALT) is frequently utilized as a TMM in human sarcoma subtypes. Histiocytic sarcoma (HS) is a rare but aggressive tumor of hematopoietic origin with unknown ALT incidence in humans. ALT has been identified in canine HS, a tumor type comparable to human HS that occurs with high rates in certain canine breeds such as Bernese mountain dogs (BMDs). This retrospective study characterized the frequency of ALT in BMD and non-BMD patients diagnosed with HS as surrogates for humans. Formalin-fixed paraffin-embedded tumor samples from 63 dogs at two centers, including 47 BMDs, were evaluated for their ALT activity and relative telomere content (TC) using a radiolabel C-circle assay (CCA). Known ALT-positive samples served as controls. CCA-positive cases were validated via FISH. Two BMD samples showed ALT activity of 1–14% compared to controls. All other samples were ALT-negative. The TC did not correlate with the CCA results. ALT positivity was validated by the appearance of ultrabright telomere foci. Low ALT activity was present in 4% of BMDs with HS and therefore does not appear to be a common target for therapeutic approaches but can have diagnostic value.

## 1. Introduction

Telomeres are specialized structures in cells located at the ends of eukaryotic chromosomes and, in vertebrates, consist of DNA hexamer sequences (TTAGGG)_n_ that are repeated many times [1]. Telomeres act as caps that protect against genetic instability but shorten with each cell division in somatic cells such as fibroblasts [1]. Progressive telomere shortening leads to replicative senescence or apoptosis, barriers that cancer cells overcome by activating telomere maintenance mechanisms (TMMs) [2]. The two known TMMs are based on telomerase and the alternative lengthening of telomeres (ALT), which are usually not activated together. Telomerase is preferentially activated in most human and canine tumors [3,4]. However, ALT is used in about 10–15% of human cancers as a telomerase-independent TMM based on homologous recombination [5,6]. The measurement of ALT activity has the potential to become a quantifiable tumor marker that is important for prognosis and diagnosis as well as for personalized therapy as TMM-specific therapeutics emerge [5,6,7]. In addition, the therapeutic inhibition of telomerase can lead to ALT as a resistance mechanism [8].

ALT has been found to be active in certain human tumor types, particularly those derived from mesenchymal tissues such as sarcomas [9,10]. In canine tumors, knowledge about the incidence and prognosis of TMMs is limited but seems to be closely related to humans. For example, in a previous study, we detected ALT activity in 10% of canine sarcomas, including histiocytic sarcoma cases [11]. We also reported that the ALT status was a negative prognostic factor for overall survival in dogs with appendicular osteosarcoma [12].

Various methods have been used to detect ALT activity in humans and dogs, including long and heterogeneous telomere lengths and the presence of ultrabright telomere foci or C-circles (CCs) [11,13,14,15]. CCs are partially single-stranded extra-chromosomal circles of telomeric DNA. The detection of ALT-specific CCs using a C-circle assay (CCA) or of ultrabright foci using telomere-specific fluorescence in situ hybridization (ALT-FISH) have become front-line ALT assays to identify ALT activity [11,13,16,17].

In human histiocytic tumors, the clinical relevance of ALT is currently unknown. One study suggested a possible association between telomerase reverse transcriptase, clinical presentation and disease progression, without finding evidence of ALT activity [18]. Histiocytic sarcomas (HSs) are rare in both species, but they are highly malignant tumors characterized by a high rate of metastasis and low overall survival [19,20,21]. HSs are a subset of dendritic and histiocytic neoplasms that arise from CD34+ stem-cell-derived histiocytes. In dogs, HSs are overrepresented in predisposed breeds such as Bernese mountain dogs (BMDs), flat-coated retrievers and Rottweilers [22,23,24]. In particular, BMDs are more likely to develop HS due to a polygenic mode of inheritance and are 17 times more likely to die from neoplastic disease than other breeds [25,26,27]. Typical locations in human and canine patients affected by solitary or disseminated disease include the lungs, lymph nodes, spleen, bone marrow and various other organs [28,29,30,31,32,33].

The prevalence of ALT activity in HS is currently unknown. The fact that HS is a very rare disease in humans would make dogs, with a high incidence of HS in certain breeds, an interesting model for comparative TMM research. Thus, the aim of this study was to evaluate the frequency of ALT and the telomeric content in dogs with spontaneous HS, including a cohort of BMDs.

## 2. Materials and Methods

### 2.1. Spontaneous Histiocytic Sarcoma Tissue Samples and Clinical Data

Formalin-fixed and paraffin-embedded (FFPE) tumor specimens of spontaneously occurring HS were archived at two institutions: Laboklin GmbH & Co. KG, Bad Kissingen, Germany (institution 1, 2014–2018), and the University of Veterinary Medicine Vienna (institution 2, 2010–2019). Institution 1 provided 42 samples of 39 Bernese mountain dogs (BMDs), and institution 2 provided 24 samples from 8 BMDs and 16 dogs of various other breeds. In total, 66 samples from 63 dogs were grouped into a BMD cohort and a non-BMD cohort (Table 1). The BMD cohort consisted of 50 samples from 47 BMDs, and the non-BMD cohort consisted of 16 samples from 16 non-BMD patients. In the BMD cohort, for one patient three samples (skin and mucous membranes) were available for examination and for another patient two samples of HS tumor (liver and spleen) were available for examination. For all tumor specimens, patient characteristics and tumor biopsy locations were available as minimum clinical information. The study was conducted in accordance with the Good Scientific Practice guidelines from the University of Veterinary Medicine based on the Austrian Agency for Research Integrity (OeAWI) guidelines (https://www.vetmeduni.ac.at/en/research/scientific-ethics-and-integrity (accessed on 8 August 2023)) and national legislation. As all samples submitted to institutions 1 and 2 were routine diagnostic submissions, there was no need to submit a request for animal testing or to obtain further approval from the ethics committee of the university. This approach was supported by a decision from the german government (RUF-55.2.2-2532-1-86-5).

### 2.2. Control Tissue Samples and Cell Lines

For the C-circle assay (CCA), two FFPE tumor samples from canine sarcomas (one HS and one osteosarcoma) with known ALT activity and two human osteosarcoma cell lines (U2OS and SAOS-2) with known ALT activity served as positive controls. The origin and handling of the cell lines have been recently described [11,12]. A sham sample without tissue or cells was processed accordingly and served as a negative and background control.

For the ALT-FISH analysis, three canine tumor samples with known ALT-positive or -negative statuses from a previous study as well as a human astrocytoma sample were used as controls (Table 2) [11,34].

### 2.3. DNA Extraction and Quantification

DNA was extracted from five to ten 10 µm thick slices of FFPE archived tumor specimens using the nexttec 1-Step DNA Isolation Kit for Tissue and Cells (nexttec Biotechnology GmbH, Hilgertshausen-Tandern, Germany) as previously described [12]. Prior to extraction, slices were dewaxed with xylene and cells were lysed overnight. DNA isolation from human control cell lines was performed with the Quick C-Circle Preparation protocol (QCP) [13]. The extracted DNA was quantified with a Qubit fluorometer (Invitrogen, Carlsbad, CA, USA). Briefly, 2 µL, corresponding to 1/50 of the nexttec and 1/25 of the QCP-extracted DNA, was analyzed with iQuant BR and high-sensitivity dsDNA Quantification Kits (GeneCopoeia, Rockville, MD, USA), which allowed the detection of dsDNA between 0.2 ng and 1000 ng. Therefore, a minimum of 5 ng or 0.1 ng/µL of extracted DNA could be measured. The DNA samples were stored at −80 °C.

### 2.4. Radiolabel C-Circle Assay and Telomeric Content

The ALT activity was determined with a radiolabel C-circle assay (CCA) [13]. The assay protocol was described in detail in our previous study [12].

In brief, 2 µL of extracted DNA was diluted in three steps and mixed with Tris buffer to a total volume of 10 µL to archive DNA quantities close to the 32 ng of input DNA. Rolling circle amplification (RCA) of C-circles (CCs) was performed with a 10 µL dilution mixed with a 10 µL master mix containing 7.5 units of phi29 DNA polymerase (New England Biolabs Inc., Ipswich, MA, USA) for phi+ CCA reactions or with a master mix containing the same volume of ddH2O to replace the enzyme for phi- CCA reactions. For CC detection, 20 µL of CCA products were transferred to a Biodyne B membrane (cat# P/N 60208, Pall Corporation Life Science, Port Washington, NY, USA), dried and cross-linked. Radioactive labeling was performed using the oligonucleotides for ctel and alu, and signals were measured using Storage Phosphor Screens (Amersham Biosciences, Little Chalfont, UK) and a Typhoon scanner (GE Healthcare, Chicago, IL, USA). The ctel and alu oligonucleotides hybridize with the telomere G strand and with the complementary strand of frequently occurring alu sequences, respectively [12].

Volume signal intensities (VSIs) were measured using ImageQuant TL 1D v8.1 (GE Healthcare, Chicago, MA, USA), and VSI measurements were divided by 3,500,000 to make the numbers easier to handle. CCA levels were given in arbitrary units (AUs) after background correction and normalization to the alu signal. Following published recommendations, AU values greater than five times the background (>0.204 AU) indicated ALT activity, and AU values less than two times the background were considered ALT-negative [13]. CCA levels between two and five times the background (2× BG and 5× BG) needed to be carefully assessed with appropriate controls to classify the ALT status.

The total telomeric content (TC) of all samples was calculated using the so-called Telo/Alu method via a dot blot analysis with a radiolabeled ctel oligonucleotide, as previously described in detail [12].

### 2.5. Telomere-Specific Fluorescence In Situ Hybridization (FISH)

Telomere signals were visualized on 5 µm sections of archived formalin-fixed paraffin-embedded (FFPE) tissue specimens as described in basic protocol 2 with minor modifications [35]. Sections were placed on Superfrost Plus slides (Epredia, Braunschweig, Germany), and the paraffin was melted for 15 min. Prior to dehydration, a 100 µg/mL RNase A treatment was performed for 1 h at RT with 200 µL of blocking solution covering each tissue slice to avoid signals for telomere transcripts and concluded with two 5 min TBST washes in Coplin jars. Denaturation and hybridization were performed with 0.5 µg/mL PNA-probe TelC-Cy3 (catalog number F1002, PNA Bio Inc., Thousand Oaks, CA, USA), and slides were sealed with a flexible mounting adhesive (Fixogum, Marabu, Tamm, Germany). Nuclei were stained with 1 µg/mL DAPI in 200 µL of TBST per slide. Then, 40 µL of Vectashield Antifade Mounting Medium (catalog number H-1000-10, Vector Laboratories Inc., Newark, CA, USA) was added before a coverslip was carefully applied to each slide and sealed with nail polish. A confocal microscope (Olympus IXplore SpinSR Spinning Disk Confocal Microscope with Yokogawa Disc, EVIDENT Corporation, Tokyo, Japan) with a 50 nm pinhole SoRa disc (Yokogawa Electric Corporation, Tokyo, Japan) and an ORCA-Fusion Digital CMOS camera (C14440-20UP, Hamamatsu Photonics K.K., Tokyo, Japan) equipped with an appropriate fluorescence filter set and a 40× objective was used for the visualization and imaging of the slides. CellSens Dimensions software (version 4.1, EVIDENT Corporation, Tokyo, Japan) with the add-ons Count & Measure, CI Deconvolution and Deep Learning was used to manually train an artificial intelligence (AI) module for the automatic detection of DAPI and Cy3 signals, nuclei and telomere foci, respectively. About 70–90% of nuclei were detected by the AI after a manual inspection of the results. Telomere foci were quantified by the software as mean gray intensity values of around 10,000 cells per case and reflected the relative telomere lengths of individual telomeres within interphase cells [17].

The presence of large, very bright intranuclear foci of telomere FISH signals in more than 1% of cells was used to classify cells as ALT-positive [35]. Briefly, the number of ultrabright intranuclear telomere foci (UBITF) was measured. UBITF show more than 10 times the mean gray level of telomere signals found in canine sarcomas lacking ALT. In addition, giant intranuclear telomere foci (GITF) were defined as areas with more than 4 times the mean area level of telomere signals found in canine sarcomas lacking ALT. These thresholds of telomere signal and size were established using a grade I soft tissue sarcoma (case ID 5682) that was found to be ALT-negative in a previous study (Table 2) [11].

### 2.6. Statistical Analysis

A statistical analysis was performed using the GraphPad Prism 5 software package (Version 5.02). The correlation between CC levels and TC was analyzed using Spearman’s correlation coefficient.

## 3. Results

### 3.1. Patient Characteristics and Clinical Data

Of the 63 dogs in this study, 47 were BMDs and 16 dogs belonged to the non-BMD group (13 pure breeds and 3 mixed breeds). Demographic patient characteristics and locations are shown in Table 1. Overall, most dogs were middle-aged, between 5 and 10 years, at the time of diagnosis (mean: 8.1 years). The tumor site was recorded in 55/63 dogs. The most prevalent tumor sites were the skin/subcutis in both cohorts, followed by disseminated disease and the spleen.

In the BMD cohort, the mean age was 7.6 years and 77% of BMDs were between 5 and 10 years. HS in the skin was most prevalent, with 40% of cases, and disseminated disease was the second most prevalent, with 17% of cases.

In the non-BMD cohort, the mean age was slightly higher at 9.5 years, and again most patients localized in the middle-aged range (63%). The skin was also the most prevalent site and with 63% of cases had a higher prevalence than the BMDs. The other sites were disseminated HS (19%), the spleen (12%) and the central nervous system (6%).

### 3.2. ALT Activity in Canine Histiocytic Sarcoma Tissue and Controls

A total of 66 tumor samples from spontaneously occurring HS cases of 63 dogs were screened for ALT activity with the radiolabel C-circle assay (Figure 1). In total, 50/66 tumor samples belonged to the BMD group (47 patients) and 16/66 samples belonged to the non-BMD group (16 patients). In two patients of the BMD group, tumor samples were available from two and three locations, respectively.

After DNA extraction, an optimal input DNA content of 32 ng could be used for CCA in 62/66 samples; in four samples, less than 32 ng was available (range: 2.1 to 30 ng). Alu probe signals corresponding to the DNA quantities of phi+ and phi- dots, as well as duplicates, showed good optical agreement. Indeed, the signals of 66 samples demonstrated a coefficient of variation of 21.8% and ruled out signal intensity variations due to inaccuracy during the assay procedure.

Low ALT activity (CCA+) with a signal intensity just above the five-fold threshold of 0.20 AU was detected in two tumor samples from BMD cases (Figure 1A). Furthermore, borderline ALT activity between the two- and five-fold thresholds was detected in two tumor samples from the non-BMD group. In CCA− samples, signals with polymerase (phi+) were similar to sample-specific background signals without polymerase (phi−).

In detail, the two weak CCA+ cases resulted in CCA values of 0.21 AU and 0.22 AU, just above the five-fold background threshold of 0.20 AU (Figure 1B). The two borderline CCA+ cases had CCA values of 0.091 AU and 0.093 AU, which were 2.2- and 2.3-fold above the background, respectively. All other tumor samples in the BMD and non-BMD cohorts were CCA-, with CC signals below the two-fold above background threshold. The CC signals ranged from −0.17 to 0.08 AU in the BMD group and from −0.80 to 0.08 AU in the non-BMD group. In two BMD patients, tumor samples were collected from two and three different tumor locations, all of which were negative (CCA-).

All positive control tumor tissues and cell lines showed CCA levels in the expected positive ranges, which was important to exclude false negatives. In detail, the known ALT canine HS had a CC signal of 3.9 AU, and the known ALT canine osteosarcoma showed a CC signal of 1.6 AU. The two human ALT cell lines reached CC signals of 6.7 AU (U2OS) and 17.9 AU (Saos-2) (Figure 1A,B). Compared to controls, the two BMDs showed ALT activity of 5.4–13.9% of the canine control tissues and 1.2–2.7% of the human control CLs. The negative background controls on each blot showed no measurable CC signals. For 9/66 tumor tissue samples (14%), technical replicates were analyzed using CCA, and the AU levels of the replicates ranged from −0.08 AU to 0.1 AU and resulted in the same negative ALT status without strong variation in the AU levels for the positive control U2OS (CV = 28%). In conclusion, ALT activity in HS was observed by CCA in 4% of BMDs and ≤9% of non-BMDs.

### 3.3. Telomeric Content in Canine Histiocytic Sarcoma Tissue and Controls

The total telomeric content (TC), measured using the radiolabel Telo/Alu method, in all canine HS samples ranged from 0.15 to 0.77 AU (median: 0.41 AU) in the BMD group and from 0.16 to 0.65 AU (median: 0.34 AU) in the non-BMD group. The two BMD cases with low ALT activity had TC values of 0.71 and 0.73 AU, which were elevated compared to the median and therefore what was found in most of the cases studied (Figure 2). The two non-BMD cases with borderline ALT activity had TC values of 0.21 and 0.24 AU, which were below the values found in most cases.

Among the positive controls, the canine HS and osteosarcoma had TC values of 0.19 AU and 0.21 AU, and the two human cell lines had TC values of 0.19 (U2OS) and 0.10 AU (SAOS-2). No correlation was found between the CCA levels and the TC of canine HS tumor tissues (Spearman r = −0.054, *p* = 0.670) (Appendix A).

### 3.4. Evaluation of CCA Results Using Ultrabright and Giant Intranuclear Telomere Foci Detection

Cases with weak positive and borderline CCA results were validated using FISH for the detection and quantification of the telomere signals in interphase nuclei of FFPE sections (Figure 3). The detection of nuclei and telomeres was performed using the signals of DAPI and Cy3, respectively. In total, 6 to 12 independent micrograph areas per case were analyzed with the help of trained AI software (add-on for Deep Learning of CellSens Dimensions software, version 4.1) that can recognize and quantify the nuclei and intranuclear telomere foci of interphase cells (Table 2). Both HS cases classified by CCA results above the five-fold background threshold as ALT-positive were confirmed by the appearance of more than 1% of nuclei with telomere UBITF signals. In contrast, the two HS cases classified as uncertain as ALT-positive by CCA results between the two- and five-fold background thresholds were not validated by FISH analyses. Published canine and human sarcoma and astrocytoma cases known for the presence or absence of ALT served as controls and resulted in the expected relative numbers of nuclei with telomere UBITF signals. In addition, the intensity of all intranuclear telomere signals normalized using DNA stain signals resulted in mean TC values via FISH analyses. ALT-positive cases validated using FISH showed a higher TC compared to cases that were not validated. The TC results using FISH were in accordance with the results obtained using the radiolabel Telo/Alu method, as described before. As ALT cells can contain telomere clusters, we measured the size of the telomere foci and the appearance of GITF in interphase cells (Table 2). However, this size parameter was not associated with the signal intensity for two of the cases studied using qFISH, one of the questionable ALT-positive cases (case ID 5332) and one of the positive controls (case ID 6455). UBITF and GITF occurred at different frequencies in cells, and less than 40% of telomere foci were UBITF and GITF in combination (Appendix A).

The two BMD cases with validated ALT activity came from cohort 1 (institution 1) and were a 14-year-old male dog with a disseminated HS and a 7-year-old female dog suffering from a cutaneous HS.

## 4. Discussion

This is the first study investigating the incidence of ALT activity in canine HS in a large cohort. Forty-seven dogs of BMD breeds were collected by two institutions and constituted a large homogenous population with HS that was evaluated together with 16 non-BMD breeds for the presence of ALT. A key finding was that ALT activity was found in only 4% of canine HS in BMD breeds and not at all in non-BMD breeds. Thus, in this studied cohort, the ALT mechanism is not a preferred TMM used by this tumor entity.

The low ALT activity contrasts with previous findings in a smaller cohort where 2/5 HS cases were detected as ALT-positive [11]. This study demonstrated that the tissue samples of the two cases showed clear ALT positivity with CC activity 21-fold and 24-fold above the threshold. Such findings could not be verified by this work using a larger number of tumor samples (66 samples) despite clear positive controls. The CCA-positive BMD cases showed rather low AU levels just above the two- and five-fold background positivity thresholds. Before cases with such low levels can be classified as ALT-positive, other methods of ALT detection should be applied for validation, as recommended [13]. We used ALT-FISH to detect the presence of ALT via the appearance of cells with ultrabright telomere foci on FFPE tumor tissue samples [17]. Indeed, only two cases above the five-fold background threshold were validated using ALT. In addition, we could show that the telomere content (TC) cannot be used to determine the TMM. This result is in accordance with previous findings [11,12].

The low prevalence of ALT in histiocytic sarcoma is surprising. In general, about 10–15% of tumors use ALT as their maintenance mechanism, but some tumors like mesenchymal tumors tend to show high ALT activity in both humans and canines [6,12,36]. For example, osteosarcomas show a prevalence over 60% in humans and a prevalence of 20% in canines. In both species, the presence of ALT seems to have an impact on prognosis [12,37,38]. But tumors are a heterogeneous group, and a lot may still be unknown, especially in rare tumor entities such as HS.

In humans, only scarce data regarding the TMM activity in histiocytic diseases are available. Despite growing data about ALT’s prevalence, to our knowledge no studies of ALT in canine or human HS have been conducted yet [36]. Only some evidence of telomerase activity exists within the extended family of histiocytic diseases [18,39]. More information about the TMMs used in HS could help in tackling this still-fatal disease in both humans and canines. Especially in humans, the very small case numbers of HS coupled with a high fatality rate result in few opportunities to study this disease and require other species, such as dogs, to enable research [27]. Targeted therapies like tyrosine kinase inhibitors or monoclonal antibodies are used in the treatment of HS with some effect [20]. ALT does not seem to be generally beneficial for new diagnostic and therapeutic approaches [20]. The telomerase activity is presumably the more commonly used TMM in this tumor entity, but this has to be confirmed. Therapeutic approaches targeting telomerase have been developed, but this therapy may cause ALT as a resistance mechanism [40]. Further investigation of the TMMs in HS is needed to draw conclusions about potential therapeutic applications.

A limitation of this study is the low number of ALT-positive cases, making further correlations impossible despite the gathering of a large number of samples from one dog breed from different institutions to avoid bias. The latter only further emphasizes the low prevalence of ALT in HS. The use of FFPE material to ensure a larger number of cases hindered some TMM analyses such as the analysis of telomerase activity. The radiolabel CCA method used is highly sensitive and reproducible for the detection of ALT in canine sarcoma [11]. However, we cannot exclude false-negative cases because only a subset of samples was run in duplicate and not all samples were screened for UBITF. Furthermore, only limited clinical data were available. Therefore, disseminated patients, especially in the BMD cohort, might be underrepresented. Nevertheless, this is of low relevance because no clinical conclusion was drawn.

In summary, the results indicate, contrary to our previous study, that HS has rare and weak ALT activity. ALT seems to be used only in isolated cases, and further investigations of the preferred TMM in canine and human HS are needed. The high occurrence of HS in BMDs can still be used to help tackle this fatal human disease. Investigation of other TMMs or molecular targets that are more frequently used is warranted and might help human and canine patients simultaneously. This makes dogs and the knowledge of the TMMs used in these tumors a special interest in developing targeted therapeutics. Canine HS might be a good model for comparative research and for the very rare cases occurring in humans.

## 5. Conclusions

ALT activity was evaluated in a total of 63 dog patients from two centers, including a homogeneous population of 47 BMDs, which are known to have a breed predisposition for HS. Only two BMD patients showed CCA+ levels above the threshold, and two non-BMD patients showed borderline CCA+ activity. ALT positivity was validated by the appearance of ultrabright telomere foci and was confirmed in the two BMD cases. This means that ALT is used only infrequently in BMDs and not at all in non-BMD patients. Therefore, 2/47 (~4%) BMDs used the ALT mechanism, which seems to be rare in canine HS compared to canine OS, where ALT is used in 20% of tumors, and ALT is useful only in a few cases to target HS [12].

## Figures and Tables

**Figure 1 cancers-15-04214-f001:**
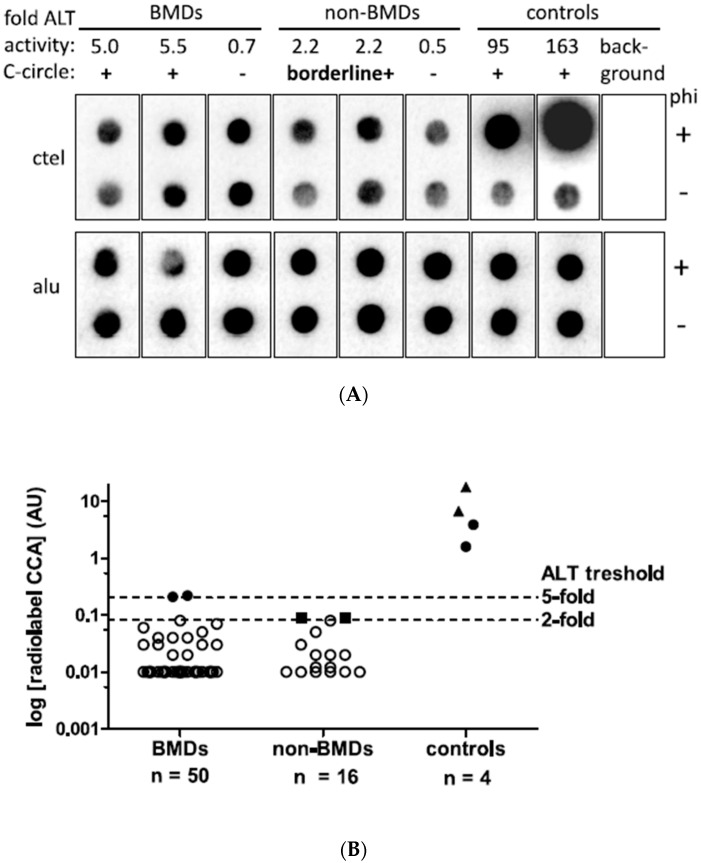
C-circle assay (CCA) results of canine histiocytic sarcoma. (**A**) Representative radiolabel CCA of canine HS samples. Dot blots including the two CCA-positive (CCA+), two borderline CCA-positive (borderline+) and representative negative samples as well as ALT-positive canine histiocytic sarcoma and human osteosarcoma controls and the background control hybridized with ctel (for C-circles) and alu (for input DNA) oligonucleotides are shown. Rolling circle amplification was performed with and without phi29 polymerase (phi+ and phi−). (**B**) CCA activity levels in arbitrary units (AU). The two ALT histiocytic sarcoma samples of BMDs and known ALT canine FFPE sarcoma samples (full dots) as well as human ALT control cell lines (CLs, full triangles) show CC levels above the 5-fold ALT positivity threshold of 0.204 AU. Two samples in the non-BMD category showed CCA levels between 2- and 5-fold above background and were considered borderline positive (full square symbol). Samples below 2-fold above background were classified as ALT-negative (open symbols). The dashed lines indicate ALT thresholds of CCA levels 2- and 5-fold above background. To plot negative data points on a logarithmic scale, negative and zero values were assigned a value of 0.01.

**Figure 2 cancers-15-04214-f002:**
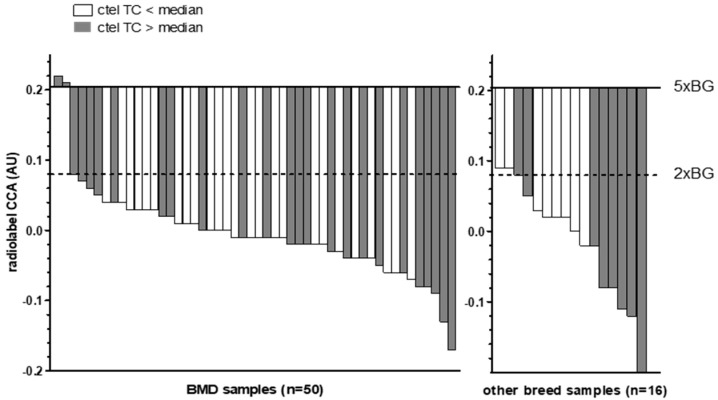
Waterfall plot of C-circle assay (CCA) results compared to median radiolabel telomeric content (TC) values. Bars show CC levels of canine HS tumor tissue samples in arbitrary units (AU). Baselines indicate the 5-fold (5× BG) and 2-fold (2× BG) thresholds for ALT positivity. Filled bars indicate samples with a radiolabel ctel TC above the median (>0.41 for BMDs and >0.34 for the non-BMDs), whereas open bars indicate a ctel TC below the median.

**Figure 3 cancers-15-04214-f003:**
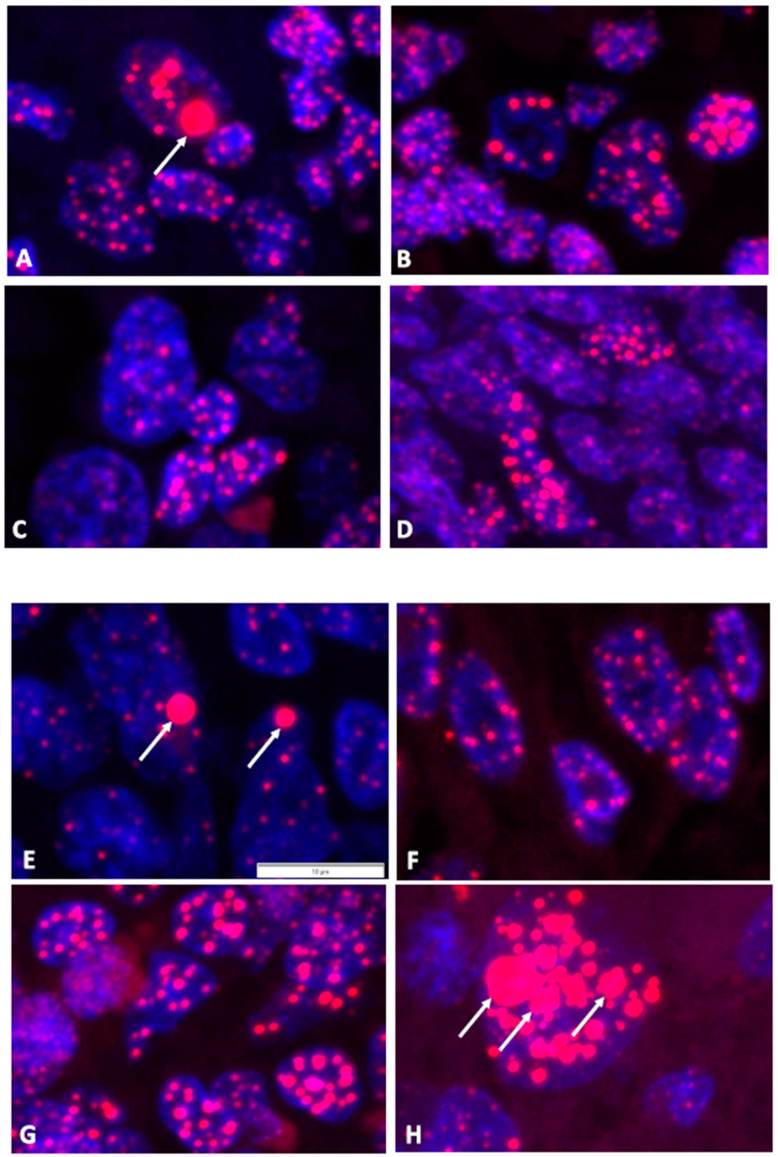
Tissue sections from FISH analyses. Micrograph examples with florescence signals of nuclei (DAPI) in blue and telomere foci (TelC-Cy3) in red. (**A**,**B**) Canine HS cases 86638 and 23671, respectively, with AU > 5× BG. (**C**,**D**) Canine HS cases 5332 and 5249, respectively, with AU = 2× BG–5× BG. (**E**) Canine HS case 89-2 as positive control for ALT. (**F**) Canine soft tissue sarcoma case 5682 as negative control for ALT. (**G**) Canine hemangiosarcoma case R6455 as positive control for ALT. (**H**) Human astrocytoma case YTBO as positive control for ALT. Arrows indicate examples of UBITF. The 10-µm scale bar is shown in (**E**) and applies to all micrographs.

**Table 1 cancers-15-04214-t001:** Patient characteristics.

Parameters	All (n)	BMDs (n)	Non-BMDs (n)
**Breeds**			
mixed breeds	3	0	3
purebred	60	47	13
**Sex**			
male/neutered *	28/13	19/10	9/3
female/spayed *	30/11	25/7	5/4
unknown	5	3	2
**Age categories**			
mean age (years)	8.1	7.6	9.5
<5 years	6	5	1
5–10 years	46	36	10
>10 years	8	4	4
unknown	3	2	1
**Tumor site**			
skin	29	19	10
disseminated	8	5	3
spleen only	11	9	2
CNS	1	0	1
lymph nodes	2	2	0
liver	2	2	0
other	2	2	0
unknown	8	8	0
**total number of cases**	63	47	16

n = number of cases; BMD = Bernese mountain dogs; * of total number of males or females.

**Table 2 cancers-15-04214-t002:** Telo qFISH results *.

			AU to BG Values of CCA			Cells Quantified	UBITF Cells		GITF Cells				
Case ID	Species/ Breed	Diagnosis	<2	2–5	>5	#	#	%	#	%	ALT	TC	Reference
86638	BMD	Histiocytic sarcoma			+	10,153	2506	24.68	2478	24.41	+	2.62	this study
23671	BMD	histiocytic sarcoma			+	14,532	2385	16.41	922	6.34	+	3.00	this study
5332	non-BMD	histiocytic sarcoma		+		12,023	21	0.17	1219	10.14	−	1.50	this study
5249	non-BMD	histiocytic sarcoma		+		19,953	3	0.02	402	2.01	−	1.57	this study
89-2	non-BMD	histiocytic sarcoma			+	13,400	1534	11.45	2817	21.02	+	1.82	[11]
5682	non-BMD	soft tissue sarcoma	+			5969	5	0.08	168	2.81	−	0.99	[11]
6455	non-BMD	hemangiosarcoma			+	12,975	6090	46.94	62	0.48	+	3.33	[11]
YTBO	human	astrocytoma			+	3283	292	8.89	263	8.01	+	1.13	[34]

* Symbols: #, number; + positive; − negative.

## Data Availability

The data presented in this study are available in this article.

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
