# Peer review of "Alternative Lengthening of Telomeres Is Rare in Canine Histiocytic Sarcoma"

_cancers, 2023, doi:10.3390/cancers15174214_

Round 1

Reviewer 1 Report

In the manuscript the authors investigate the frequency of the telomere maintenance mechanism (TMM) known as alternative lengthening of telomeres (ALT) in canine histiocytic sarcoma. The authors use two well established techniques of C-circle assay (CCA) and ultra bright foci (UBF) for telomeres to identify ALT-positive samples. Out of the 63 dog patients test, only two of the samples were positive for ALT activity in both the CCA and UFB assay. The frequency of ALT positive samples reported here is lower than the 10-15% that is usually observed in other cancer types. While it is possible that the prevalence of ALT in histiocytic sarcoma in both humans and canines may in fact be lower compared to other tumors, it is hard to draw this kind of conclusion due to the limited number of samples. This may be an interesting avenue for future research as this knowledge may be important for determining the best treatment of these patients.

 The following comments or questions should be addressed by the authors:

 What does 5BG stand for in the values for CCA? If it is background this should be clear described either in the methods or figure legend.

 The terms “ctel” and “alu” to describe the radioactive probes should be briefly defined.

 In Figure 1, the dots from some of the sample seem to have poor/uneven hybridization of the radioactive probes. The results section seems to suggest that these samples were run as duplicates. If this is the case the second blots should be included or both blots should be quantified and graph so we can get a better idea on the variation. Visually, it is hard to tell that there is at least a five-fold change in the relative ctel signal after incubation with the phi29 polymerase for the two BMD samples.

 In Figure 2, there is a typo. In the chart it reads “gtel” when it should be ctel as described in the methods and figure legend.

 When testing the UBF for telomeres in interphase cells, presumably you have the actual data scored by the computer for each sample. It would be beneficial to either give both the number of total cells tested and UBF positive cells for each sample, or to have a bar graph so you can visualize the variation for each sample. You could either present the data as the percentage of cells with UBF or as the average number of UBF per cell. How does the # of UBF in the new BMD or non-BMD samples compared to the positive control HS case 89-2?

 Can the software/algorithm developed for detection of UBF for telomeres in interphase cells be adapter to measure size not just intensity? ALT cells are known to show clustering of multiple telomeres, leading to large and ultra bright foci. I am wondering if measuring “larger than normal” foci may be a more sensitive method for detection of ALT activity.

The manuscript was written well with little to no errors or awkward sentences.

Author Response

We thank the reviewer for the valuable comments to improve the work. Notes to Reviewers is written in italics below comments and changes in the manuscript have been marked by track-changes function.

The following comments or questions should be addressed by the authors:

 What does 5BG stand for in the values for CCA? If it is background this should be clear described either in the methods or figure legend.

This term and 2BG is used several times with different designations in the manuscript: in Table2, Legends to Figs. 2 and 3. We changed all terms in the manuscript to the designations 5xBG and 2xBG, and defined the terms in the methods section 2.4.

The terms “ctel” and “alu” to describe the radioactive probes should be briefly defined.

Terms “ctel” and “alu” were defined in section 2.4:
The ctel and alu oligonucleotides hybridize with the telomere G strand and with the complementary strand of frequently occurring alu sequences, respectively [12].

 In Figure 1, the dots from some of the sample seem to have poor/uneven hybridization of the radioactive probes. The results section seems to suggest that these samples were run as duplicates. If this is the case the second blots should be included or both blots should be quantified and graph so we can get a better idea on the variation. Visually, it is hard to tell that there is at least a five-fold change in the relative ctel signal after incubation with the phi29 polymerase for the two BMD samples.

We run replicates from random chosen 9 tumor tissue samples, corresponding to 9 of 66 cases (14%). Unfortunately, the samples with CCA signals above 5xBG and 2xBG were not replicated and thus cannot be included. The nine duplicated samples showed all signals below 2xBG. The variation coefficient of duplicated positive control U2OS samples was 28% and this information was included at the end of result section 3.2. We agree that it is difficult to detect volume signal intensities that are only 5 times above background with the naked eye, but this prompted us to examine these cases with a different methodology such as FISH for UBF.

 In Figure 2, there is a typo. In the chart it reads “gtel” when it should be ctel as described in the methods and figure legend.

Thank you, we corrected this.

 When testing the UBF for telomeres in interphase cells, presumably you have the actual data scored by the computer for each sample. It would be beneficial to either give both the number of total cells tested and UBF positive cells for each sample, or to have a bar graph so you can visualize the variation for each sample. You could either present the data as the percentage of cells with UBF or as the average number of UBF per cell. How does the # of UBF in the new BMD or non-BMD samples compared to the positive control HS case 89-2?

Table 2 does contain this information; we reworded the column headings to increase readability and added the breed group for canine species. We added the average number of ultrabright intranuclear telomere foci (UBITF) per cell. The UBITF analyses was performed with four cases only in addition to ALT positive and negative cases from previous studies (Table 2). Thus, the number of cells with UBITF between BMD or non-BMD samples can be compared to the positive control HS case 89-2.

 Can the software/algorithm developed for detection of UBF for telomeres in interphase cells be adapter to measure size not just intensity? ALT cells are known to show clustering of multiple telomeres, leading to large and ultra bright foci. I am wondering if measuring “larger than normal” foci may be a more sensitive method for detection of ALT activity.

Thank you for this comment. Yes, the software does calculate several additional parameter such as the size of the telomere foci. We reanalyzed the data for size and included the information in Table 2. We defined giant intranuclear telomere foci (GITF) as area more than 4-fold the mean area level of telomere signals found in canine sarcoma lacking ALT. However, this size parameter is not associated with the signal intensity for two of the cases studied by qFISH, one of the questionable ALT positive cases with CCA levels between 2 to 5xBG (Case ID 5332) and one of the positive controls (Case ID 6455). We included this information in method section 2.5, result section 3.4.

Comments on the Quality of English Language

The manuscript was written well with little to no errors or awkward sentences.

Thank you.

Reviewer 2 Report

In this study, Kreilmeier-Berger et al. analyze the prevalence of the ALT pathway in histiocytic sarcoma, a tumor type that is present in high frequency in Bernese Mountain Dogs (BMD).  Sarcomas frequently are associated with the ALT rather than the telomerase pathway. The authors take a retrospective approach to analyze samples collected from two institutions by isolating DNA and analyzing the frequency of C-circles, a product of the ALT process coupled with positive and negative controls. The authors find that C-circles are present in only 4% of BMD samples. Positives were confirmed by ultra-bright foci, also typical of ALT. Hence, the authors conclude that the ALT pathway is not frequently used in HS in BMDs.

The result was unexpected given the tumor type, and the negative outcomes will likely interest investigators studying these sarcomas. However, there are several issues to increase the rigor of the analysis.

Experimental:

1.     Since this result is negative, the authors should test all samples for ultrabright foci. The expectation would be that none of the negative samples for the circle assay would display such foci. This is critical since it is the only means of validating the conclusion, given the unavailability of telomerase assays in these fixed tissues.

2.     Technical replicates should be performed for all samples, not just a subset, to ensure the reproducibility of the results.

Textual

3.     Please site the approval documentation more thoroughly, including references for best practices and the location of the approval site (University or municipality).

4.     Spacing errors in Table 2 must be fixed.

5.     The first two sentences regarding the function of telomeres should be rewritten for clarity.

The initial description of telomeres is poorly written.

Author Response

The result was unexpected given the tumor type, and the negative outcomes will likely interest investigators studying these sarcomas. However, there are several issues to increase the rigor of the analysis.

We thank the reviewer for the valuable comments to improve the work. Notes to Reviewer is written in italics below comments and changes in the manuscript have been marked by track-changes function.

Experimental:

  1. Since this result is negative, the authors should test all samples for ultrabright foci. The expectation would be that none of the negative samples for the circle assay would display such foci. This is critical since it is the only means of validating the conclusion, given the unavailability of telomerase assays in these fixed tissues.

Thank you for this important issue. Results from your first published ALT study in canine sarcoma (https://pubmed.ncbi.nlm.nih.gov/27585244/) and from others studying human sarcoma (e.g. https://pubmed.ncbi.nlm.nih.gov/23260199/) indicate the existence of sarcoma cases using both ALT and TA, thus identification of TA if possible would not exclude ALT. In this first study, we compared several methods and found the radiolabel CCA method as most sensitive and accurate for screening and detection of ALT in canine sarcoma. In contrast, we used the more elaborate UBITF assay only to validate the few weak ALT positive cases identified by CCA together with known ALT positive and negative controls from our first study (Table 2). We understand the point taken by the reviewer but study of all samples for UBITF is currently not possible due to limited resources. Therefore, we cannot completely exclude false-negative cases. We included this limitation of our study in the discussion (page 10).

  1. Technical replicates should be performed for all samples, not just a subset, to ensure the reproducibility of the results.

The DNA amount isolated from FFPE did not allow to perform replicated experiments for all samples. Instead, a random subset of samples was chosen to evaluate the reproducibility of the results. All replicated samples confirmed the negative CCA results. In previous studies, we demonstrated the reproducibility of the CCA to detect ALT activity for canine sarcoma that include tumor tissue samples from histocytic sarcoma with and without ALT activity (https://pubmed.ncbi.nlm.nih.gov/27585244/). In detail, we analyzed 20 of 20 snap-frozen and 19 of 49 FFPE tumor tissue samples as technical and biological replicates and concluded that the CCA levels of replicates were identical in respect to CC content and showed no heterogeneity for ALT. This finding for canine sarcoma does confirm that the radiolabel CCA is a very robust method. We included this limitation of our result in the manuscript (page 10).

Textual

  1. Please site the approval documentation more thoroughly, including references for best practices and the location of the approval site (University or municipality).

References and location of the approval site have been added in methods section 2.1

  1. Spacing errors in Table 2 must be fixed.

Structure of Table 2 was fixed and additional information requested by another reviewer was included.

  1. The first two sentences regarding the function of telomeres should be rewritten for clarity.

We adapted the first two sentences:
Telomeres are specialized structures in cells located at the ends of eukaryotic chromosomes and, in vertebrates, consist of DNA hexamer sequences (TTAGGG)n repeated many times. Telomeres act as caps that protect against genetic instability, but shorten with each cell division in somatic cells such as fibroblasts [1].

Comments on the Quality of English Language

The initial description of telomeres is poorly written.

This part was rewritten as requested.

Round 2

Reviewer 2 Report

All issues have been adequately addressed.

Minor editing is required.

Author Response

The manuscript has been proofread for English language and track changes are marked.